ecology

marine cave, MIG-seq, *Parhippolyte misticia*, larval dispersal

**Author for correspondence:**
Takefumi Yorisue
e-mail: yorisue@gmail.com

# Extensive gene flow among populations of the cavernicolous shrimp at the northernmost distribution margin in the Ryukyu Islands, Japan

Takefumi Yorisue[1,2,3,4], Akira Iguchi[2], Nina Yasuda[5], Masaru Mizuyama[6], Yuki Yoshioka[7], Aika Miyagi[7] and Yoshihisa Fujita[8]

[1]Integrative Aquatic Biology, Onagawa Field Center, Graduate School of Agricultural Science, Tohoku University, 3-1 Mukai, Konori-hama, Onagawa, Oshika, Miyagi 986-2242, Japan
[2]Geological Survey of Japan, National Institute of Advanced Industrial Science and Technology (AIST), AIST Tsukuba Central 7, 1-1-1 Higashi, Tsukuba, Ibaraki 305-8567, Japan
[3]Institute of Natural and Environmental Sciences, University of Hyogo, 6 Yayoigaoka, Sanda, Hyogo 669-1546, Japan
[4]Division of Nature and Environmental Management, Museum of Nature and Human Activities, Hyogo, 6 Yayoigaoka, Sanda, Hyogo 669-1546, Japan
[5]Department of Marine Biology and Environmental Science, Faculty of Agriculture, University of Miyazaki, Gakuenkibana-dai Nishi 1-1, Miyazaki 889-2192, Japan
[6]Graduate School of Engineering and Science, University of the Ryukyus, 1 Senbaru, Nishihara, Okinawa 903-0213, Japan
[7]Department of Bioresources Engineering, National Institute of Technology, Okinawa College, 905, Henoko, Nago, Okinawa 905-2192, Japan
[8]General Educational Center, Okinawa Prefectural University of Arts, 1-4, Shuri Tounokura-cho, Naha-shi, Okinawa 903-8602, Japan

 TY, 0000-0003-0320-0350; AI, 0000-0002-8894-1977

Marine cave habitats in the Ryukyu Islands, Indo-West Pacific, are located at the northern edge of the distribution of many cave-dwelling species. At distribution margins, gene flow is often more restricted than that among core populations due to the smaller effective population size. Here, we used high-throughput sequencing technology to investigate the gene flow pattern among three sampling sites of a marine cave-dwelling species at the margin of its distribution range. We collected individuals of the barbouriid shrimp *Parhippolyte misticia* from three marine caves in the Ryukyu Islands and performed population genetic analyses by means of multiplexed inter-simple sequence repeat genotyping by sequencing. Based on 62 single-nucleotide

polymorphism markers, no clear population structure or directional gene flow pattern was found among the three sites. These results were unexpected because previous studies of other stygobitic shrimps in this region did find significant population genetic structures and northward directional gene flow patterns. Together, these inconsistent findings imply that marine cave-dwelling species in the region have different mechanisms of larval dispersal. Future studies on larval ecology and the biotic and abiotic factors influencing gene flow patterns are needed to clarify the mechanisms underlying the population dynamics of marine cave-dwelling species.

# 1. Introduction

Populations of coastal organisms have declined due to recent anthropogenic pressures such as coastal development, pollution and over-harvesting (e.g. [1–3]). To conserve coastal organisms, understanding gene flow patterns is important as it contributes to designing effective marine protected areas (e.g. [4,5]).

Marine cave environments have restricted exposure to the open sea and are sometimes connected to anchialine environments, which are defined as tidally influenced subterranean estuaries [6,7]. Such marine cave environments are often characterized by highly discrete species distributions [8,9]. For example, marine cave environments provide habitats for cavernicolous fauna that are dominated by stygobitic crustaceans [10–12]. Considering the unique environments and isolated species distributions of marine caves, it is expected that connectivity between populations of stygobitic species is restricted.

Previous studies using mitochondrial DNA (mtDNA) markers have shown that many cavernicolous species show a population genetic structure that spans only a few kilometres [13–19]. However, similar studies of species that inhabit marine or anchialine caves have detected no population genetic structures that span hundreds of kilometres, suggesting high levels of gene flow via larval dispersal in these species [15,20,21]. Recently, researchers recognized that using next-generation sequencing approaches such as multiplexed inter-simple sequence repeat (ISSR) genotyping by sequencing (MIG-seq) [22] to analyse a large number of genome-wide markers is a powerful means of examining population genetic structures at high spatio-temporal resolution [23,24].

For many cavernicolous species distributed in the Indo-West Pacific, the northernmost margin of their distribution is the Ryukyu Islands, Japan (e.g. [12,25]), where it is expected that the gene flow among populations is lower than that of core populations in tropical areas due to their small effective population size (central–marginal hypothesis) (e.g. [26]). It is, therefore, important to elucidate gene flow patterns at the distribution margin to design effective marine protected areas within the marginal region. Population genetic analyses using cytochrome c oxidase subunit I (COI) sequences in anchialine shrimp species in this region have revealed the presence of significant population genetic structures [18,19] and a northward directional gene flow [27]. Further analyses using genetic markers with greater spatio-temporal resolution than that afforded by COI sequences will allow us to elucidate in more detail the population genetic structures and gene flow patterns of these species in the distribution margin [28].

*Parhippolyte misticia*, a barbouriid shrimp, inhabits marine and anchialine caves in Palau [29], Papua New Guinea [30,31] and the Ryukyu Islands [31,32]. In the Ryukyu Islands, *P. misticia* is a dominant benthic species in marine caves. It is therefore a suitable species with which to investigate the gene flow pattern of cavernicolous species in the Ryukyu Islands. This is the first MIG-seq-based population genetic analysis reported for a cavernicolous species distributed in a northern peripheral population.

# 2. Material and methods

## 2.1. Sample collection

We conducted SCUBA sampling at marine caves on Okinawa Island (Hedo Cave) (26°51′53.74″ N, 128°14′43.93″ E), Ie (26°43′26.31″ N, 127°49′54.86″ E) and Shimoji (24°49′22.51″ N, 125°08′07.84″ E) Islands located in the Ryukyu Islands, Japan (figure 1) in 2016–2017. We collected 24 *P. misticia* individuals in each of Hedo and Ie, and 20 individuals in Shimoji for this study. We morphologically identified *P. misticia*, and all of the samples were fixed and preserved in 99.5% ethanol.

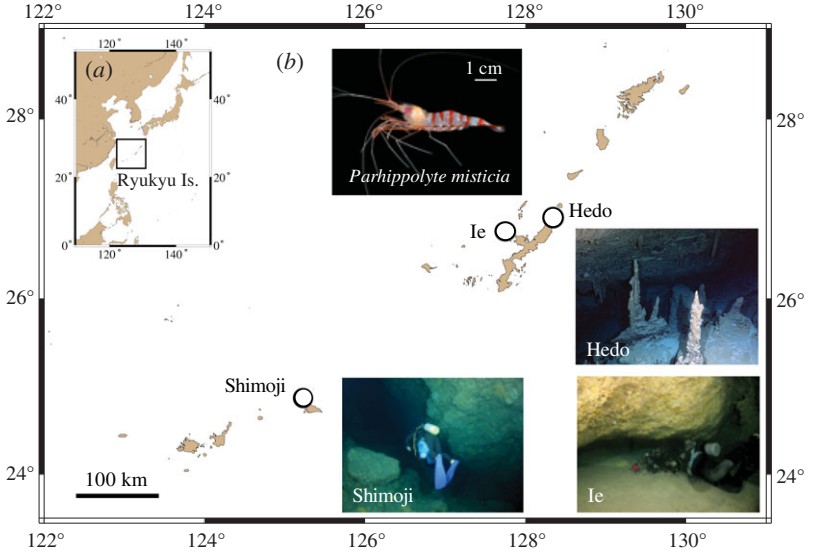

**Figure 1.** Map and photos showing sampling localities in the Ryukyu Islands. (*a*) A map of east Asia. (*b*) A map and photos of sampling location of the present study in the Ryukyu Islands, Japan. Upper photo shows *P. misticia*.

## 2.2. DNA extraction and multiplexed ISSR genotyping by sequencing (MIG-seq)

DNA was extracted from the tissue samples by using a DNeasy Blood & Tissue Kit (Qiagen, Hilden, Germany). Genome-wide single-nucleotide polymorphisms (SNPs) were obtained by using the protocol of [22]. In brief, multiplexed ISSR genotyping by sequencing (MIG-seq) was used to amplify a few hundred to a few thousand genome-wide SNPs around ISSR regions by using eight universal pairs of multiplex ISSR primers (MIG-seq primer set 1) for the first PCR. Then, DNA libraries with different indices were pooled and sequenced using MiSeq (MiSeq Control Software v. 2.0.12, Illumina) and a MiSeq Reagent v. 3 150-cycle Kit (Illumina). A total of 68 *P. misticia* individuals were included in the MIG-seq analysis. To eliminate low-quality reads and primer sequence reads from the raw data, we used the FASTQ Quality Filter tool in FASTX-Toolkit v. 0.0.14 (http://hannonlab.cshl.edu/fastx_toolkit/index.html) with a setting of −Q 33 −q 30 −p 40. Adapter sequences were removed for the MiSeq run from both the 5′ (GTCAGATCGGAAGAGCACACGTCTGAACTCCAGTCAC) and 3′ (CAGA-GATCGGAAGAGCGT-CGTGTAGGGAAAGAC) ends by using Cutadapt v. 1.13 [33], and then short reads less than 80 bp were excluded. The quality-filtered sequences were demultiplexed and filtered through Stacks v. 1.46 [34,35]. Stacks v. 1.4 [35] was used to stack the reads and extract SNPs: first, we used the ustacks program with the option settings of 'minimum depth of coverage required to create a stack (*m*)' = 10, 'maximum distance allowed between stacks (*M*)' = 1 and 'maximum distance allowed to align secondary reads to primary stacks (*N*)' = 1, and the deleveraging (d) and removal (r) algorithms were enabled; we then used the cstacks program with the option setting of 'number of mismatches allowed between sample loci when build the catalogue (*n*)' = 4 followed by sstacks. We used the population software implemented in Stacks v. 1.4 with the minimum percentage of individuals required to process a locus across all data (r) set at 70% and the data analysis restricted to a single SNP per locus. Raw sequences data have been deposited in the DNA Data Bank of Japan (DDBJ) databases with accession no. DRA008872.

## 2.3. Population genetic analyses

In total, we detected 5454 loci that matched to the catalogue, which resulted in 62 SNP markers from filtering method (i) and (ii), respectively (see DNA extraction and MIG-seq). We used BayeScan v. 2.0 [36] to detect possible SNPs under natural selection with a default setting.

A Bayesian individual-based assignment approach as implemented in STRUCTURE 2.3.4 [37] was used to examine the genetic boundaries among the three sampling sites. Twenty independent runs were performed in STRUCTURE using an admixture model and allele frequency-correlated model. Length of the burn-in period was 200 000 and the number of Markov chain Monte Carlo analyses (MCMC) was 1 000 000. We estimated ΔK [38], the most likely number of clusters using STRUCTURE HARVESTER [39]. CLUMPAK [40] was used to summarize and visualize the STRUCTURE results.

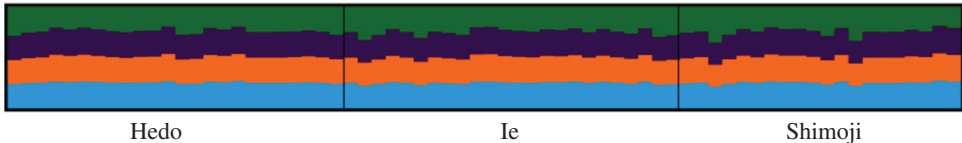

| | Hedo | Ie | Shimoji |

**Figure 2.** Result of STRUCTURE analysis. $K = 4$ with Mean (LnProb) $= -1284.375$ and Mean (similarity score) among 20 runs $= 0.998$ using 62 SNPs.

**Table 1.** *Parhippolyte misticia* pairwise population $F_{ST}$ values (below diagonal) and $p$-values (above diagonal) calculated via AMOVA.

| | Hedo | Ie | Shimoji |
|---|---|---|---|
| Hedo | | 0.003 | 0.006 |
| Ie | 0.060 | | 0.038 |
| Shimoji | 0.047 | 0.015 | |

The $p$-values were adjusted for multiple comparisons by using false discovery rate correction.

**Table 2.** Summary of genetic diversity of *Parhippolyte misticia*, as determined by MIG-seq analysis.

| | $N_a$ | $H_O$ | $H_E$ | $F$ |
|---|---|---|---|---|
| Hedo | 1.500 (0.064) | 0.109 (0.028) | 0.091 (0.018) | −0.058 (0.050) |
| Ie | 1.629 (0.052) | 0.130 (0.030) | 0.105 (0.018) | −0.013 (0.057) |
| Shimoji | 1.677 (0.060) | 0.112 (0.028) | 0.102 (0.017) | 0.076 (0.062) |

$N_a$, Number of observed alleles; $H_O$, observed heterozygosity; $H_E$, expected heterozygosity; $F$, fixation index. Values in parentheses are standard errors.

Mean number of observed alleles ($N_a$), observed heterozygosity ($H_O$), expected heterozygosity ($H_E$) and fixation index ($F$) were estimated by using GenAlEx v. 6.5 [41]. Pairwise $F_{ST}$ values were estimated using AMOVA in GenAlEx v. 6.5. Statistical significance levels for all pairwise tests were 0.05 after adjusting for multiple comparisons by using false discovery rate correction [42]. In addition, the gene flow pattern among sampling sites was estimated by using divMigrate online [43] to obtain a relative migration network graph with relative values of gene flow among sampling sites, scaled to the largest magnitude estimated. We used $N_m$ as a measure of genetic distance. The significance of asymmetric gene flow among sampling sites was tested by using 1000 bootstrap iterations.

# 3. Results

In total, we detected 5454 loci that matched to the catalogue, which resulted in 62 SNPs that are available for the population genetic analyses. BayeScan indicated that all the loci were neutral ($q$-values $> 0.05$). STRUCTURE indicated no genetic differences among the three sampling sites (figure 2). $\Delta K$ of the STRUCTURE results indicated that $K = 4$ best explained the data with a mean likelihood $= -1284.375$. Mean similarity score among 20 independent runs was 0.998. Pairwise $F_{ST}$ values among three sites were low (0.015–0.060; table 1). Number of observed alleles ($N_a$), observed heterozygosity ($H_O$) and expected heterozygosity ($H_E$) were comparable among the sites (table 2). divMIGRATE analysis is shown in figure 3, and strong gene flow with no asymmetric gene flow pattern was detected among the sites ($p > 0.05$).

# 4. Discussion

Contrary to the present results that indicated genetic homogeneity among three sites, significant population genetic structures have been detected in various stygobitic species, even at the fine

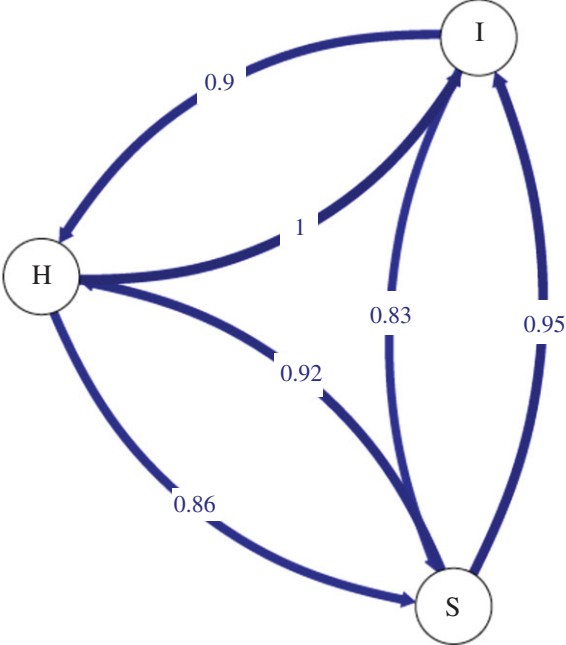

**Figure 3.** Directional relative migration networks of *Parhippolyte misticia* constructed with divMigrate using Nm. H, Hedo Cave; I, Ie Island; S, Shimoji Island.

scale [13–19]. It has been suggested that the mode of larval development can affect gene flow patterns among marine or anchialine cave habitats [21]. In general, planktotrophic larvae have a longer larval period than lecithotrophic larvae, which is advantageous for long-distance dispersal [44]. Two marine and anchialine cave species that produce planktotrophic larvae, the neritiliid snail *Neritilia cavernicola* and the alpheid shrimp *Metabetaeus lohena*, have been shown to have no significant population genetic structure over a range of approximately 200 km in the Philippines and Hawaiian Islands, respectively [20,21]. By contrast, the anchialine atyid shrimp species *Caridina rubella* and *Halocaridinides trigonophthalma*, whose larvae are also planktotrophic, have significant population genetic structures at ranges of approximately 15 km in the Ryukyu Islands [18]. *Parhippolyte misticia* also produces planktotrophic larvae (Fujita 2016, personal observation), and in the present study, we found a high level of gene flow via larval dispersal among the three sampling sites in the Ryukyu Islands. Thus, the combination of a planktotrophic mode of larval development and repeated reproduction throughout the year [31] may contribute to the high level of gene flow found in *P. misticia*.

Although northward directional gene flow has been suggested in *C. rubella* and *H. trigonophthalma* in the present study region [27], our divMIGRATE analysis did not reveal any directional gene flow among the three sites in *P. misticia*. It has been suggested that strong local-scale oceanic currents and the coastal retention of larvae can limit gene flow in *C. rubella* [18]. Therefore, if patterns of larval retention near coastal areas differ among species, this could explain these contrasting gene flow patterns. Comparative analyses of larval ecology and determination of the biotic and abiotic factors that influence gene flow patterns in marine and anchialine cave species are needed to further understand the mechanisms underlying the population dynamics of marine cave-dwelling species.

Ethics. We followed a guideline of Okinawa Prefecture, Japan to collect *Parhippolyte misticia* individuals.

Data accessibility. Raw sequence data have been deposited in DNA Data Bank of Japan (DDBJ) with accession no. DRA008872.

Authors' contributions. A.I. and Y.F. conceived the study. T.Y. and A.I. analysed data, and N.Y., Y.Y. and A.M. contributed to molecular experiments. M.M. and Y.F. contributed to field survey. T.Y. wrote the first draft of the manuscript, and A.I., Y.F., N.Y., Y.Y., M.M. and A.M. authored or reviewed drafts of the paper and approved the final draft.

Competing interests. The authors declare no competing interests.

Funding. This work was supported by the Japan Society for the Promotion of Science (JSPS) under KAKENHI grants 17H04996 to N.Y., and 16K07490, 16H06309 and 20H03313 to Y.F.

Acknowledgements. We thank two anonymous reviewers who gave useful comments to improve the original manuscript.

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
