## [Reviewer comments · Royal Society Open Science]

Review History

RSOS-191731.R0 (Original submission)

Review form: Reviewer 1

Is the manuscript scientifically sound in its present form?

No

Are the interpretations and conclusions justified by the results?

No

Is the language acceptable?

Yes

Do you have any ethical concerns with this paper?

No

Have you any concerns about statistical analyses in this paper?

Yes

Recommendation?

Major revision is needed (please make suggestions in comments)

Comments to the Author(s)

Review of manuscript RSOS-191731 entitled “Extensive gene flow among populations of the cavernicolous shrimp at the northernmost distribution margin in the Ryukyu Islands, Japan” by Yorisue et al.

The manuscript by Yorisue et al., attempts to investigate patterns of gene flow of the anchialine shrimp *Parhippolyte misticia* at its distribution margin using high through-put sequencing technologies (i.e. ISSR genotyping). The authors sampled 3 locations and found no population structure and evidence of high geneflow between all sampling locations. While I found the subject matter to be interesting, I feel that the authors failed to provide enough details to justify the merit of their work. The impact of this study could be greatly increased if more sampling sites were included. However, if that is beyond the means of this study, providing more detail throughout the manuscript would improve it greatly.

While I believe this manuscript would benefit from heavy revisions before an in-depth review, some of my major comments are outlined below.

Introduction:

Line 47- delete “therefore”

Not sure I would agree with Line 58-59. mtDNA can be useful in examining larval dispersal potential, but agree MIG-seq should provide much better resolution.

If *P. misticia* is so common in marine caves and is the dominant benthic species in the anchialine environment, the why were more sites and individuals analyzed? Is *P. misticia* really the most dominant species in the anchialine caves of the Ryukyus? The scope and impact of this work would be greatly increased by adding in more sampling sites. It might be beyond the scope of this study, but if you could include sites from the center of the distribution you could make comparisons regarding the population structure and gene flow at the center vs the margin of the distribution.

Material and Methods:

-More information regarding the sampling is warranted. In total, 48 individuals from three sampling sites were included in the MIG-seq analysis. Was this 16 individuals from each site or how was it divided up? Was that all that was collected or why were only 48 individuals analyzed?

-Please provide more detail on SNP filtering. The authors mention that “MIG-seq was used to amplify a few hundred to a few thousand genome SNPs”, but they only end up analyzing 460 SNPs. Why were more SNPs not included.

-References in the “Population genetic analyses” (and throughout the manuscript) are improperly formatted. Each reference is given twice. Once as a number corresponding to the references list and once as the author and year of publication.

Results:

Please provide more details in the results section. You have generated a lot of data with the MIG-seq analysis, however not much detail is provided. Overall, how many SNPs were found? Were there SNPs that were unique to a specific sampling site? The results of the SNP filtering? Can you use the SNPs to calculate effective population sizes? Etc.

-Please provide the results of the divMigrate analysis. You could provide the network graph or at least the relative values of gene flow between sampling sites.

-Did the authors look for F_{st} outlier SNPs?

Discussion:

In the discussion and throughout the manuscript, don't refer to the three sampling sites as “populations”. You found that there was high geneflow and no structure between the three sampling sites, so they represent one panmictic population. Instead, refer to the three sampling sites as sampling sites.

Figure 1- Add a scale bar

Table 2- What is the ratio of the number of observed alleles? Why not report just the number of alleles observed?

Review form: Reviewer 2

Is the manuscript scientifically sound in its present form?

No

Are the interpretations and conclusions justified by the results?

No

Is the language acceptable?

Yes

Do you have any ethical concerns with this paper?

No

Have you any concerns about statistical analyses in this paper?

Yes

Recommendation?

Major revision is needed (please make suggestions in comments)

Comments to the Author(s)

This study examines gene flow in a marine cave-dwelling species of shrimp in the Ryukyu Islands in the Indo-West Pacific Ocean using next-generation SNP approach. The authors found no genetic differentiation or evidence of direction gene flow from three sites sampled, which contrasts other studies of cave-obligate shrimp species in the region. This is an interesting study; however, there are several comments as outlined below that I would like to see addressed before acceptance.

Given the audience of the journal, I feel that the authors should consider broadening the focus of the Introduction for a more general audience. As written in its current form, the manuscript is very marine cave specific. There is a large body of literature on geographic patterns of gene flow in marine and many other organisms, which can be reference to better place the study in a broader context.

Do you consider *P. misticia* a cave-obligate species?

Authors should consider a supplemental table describing field sampling sites in more detail.

Were specimens of *Parhippolyte misticia* identified morphologically? Or were sites that were sampled all from previously identified populations?

Why was such a low depth of coverage (~ 3) in your stacks analysis? If you increased to a minimum depth of 10, what effect would this have on your dataset and downstream analyses?

How did you assess allele dropout?

Did you test for violations of Hardy-Weinberg equilibrium and test of linkage between any SNPs?

You might also consider testing for population structure using approaches like ADMIXTURE or fastSTRUCTURE. You could also further explore population structure using DAPC and also demographic modeling and/or partial Mantel tests (there are issues with the latter) to explore isolation-by-distance.

Line 94 – You might consider briefly describing MIG-seq in detail for those unfamiliar with this approach.

Figure 1 could also use a scale bar.

Decision letter (RSOS-191731.R0)

05-Dec-2019

Dear Dr Yorisue,

The editors assigned to your paper ("Extensive gene flow among populations of the cavernicolous shrimp at the northernmost distribution margin in the Ryukyu Islands, Japan") have now received comments from reviewers. We would like you to revise your paper in accordance with the referee and Associate Editor suggestions which can be found below (not including confidential reports to the Editor). Please note this decision does not guarantee eventual acceptance.

Please submit a copy of your revised paper before 28-Dec-2019. Please note that the revision deadline will expire at 00.00am on this date. If we do not hear from you within this time then it will be assumed that the paper has been withdrawn. In exceptional circumstances, extensions may be possible if agreed with the Editorial Office in advance. We do not allow multiple rounds of revision so we urge you to make every effort to fully address all of the comments at this stage. If deemed necessary by the Editors, your manuscript will be sent back to one or more of the original reviewers for assessment. If the original reviewers are not available, we may invite new reviewers.

- Data accessibility

If you wish to submit your supporting data or code to Dryad (<http://datadryad.org/>), or modify your current submission to dryad, please use the following link:
<http://datadryad.org/submit?journalID=RSOS&manu=RSOS-191731>

- Competing interests

- Authors' contributions

- Acknowledgements

- Funding statement

Best regards,

on behalf of Dr Michael Tobler (Associate Editor) and Kevin Padian (Subject Editor)

Associate Editor's comments (Dr Michael Tobler):

We have received feedback from two reviewers that agreed on the merit of this study. Nonetheless, they highlighted several concerns with the scope, methods, and presentation of the manuscript. Based on the reviewer feedback, a rigorously revised manuscript following all suggestions may be suitable for publication in RSOS, even if sampling of additional individuals and populations is not possible.

Reviewers' Comments to Author:

Reviewer: 1

Comments to the Author(s)

Review of manuscript RSOS-191731 entitled "Extensive gene flow among populations of the cavernicolous shrimp at the northernmost distribution margin in the Ryukyu Islands, Japan" by Yorisue et al.

The manuscript by Yorisue et al., attempts to investigate patterns of gene flow of the anchialine shrimp *Parhippolyte misticia* at its distribution margin using high through-put sequencing technologies (i.e. ISSR genotyping). The authors sampled 3 locations and found no population structure and evidence of high geneflow between all sampling locations. While I found the subject matter to be interesting, I feel that the authors failed to provide enough details to justify the merit of their work. The impact of this study could be greatly increased if more sampling sites were included. However, if that is beyond the means of this study, providing more detail throughout the manuscript would improve it greatly.

While I believe this manuscript would benefit from heavy revisions before an in-depth review, some of my major comments are outlined below.

Introduction:

Line 47- delete "therefore"

Not sure I would agree with Line 58-59. mtDNA can be useful in examining larval dispersal potential, but agree MIG-seq should provide much better resolution.

If *P. misticia* is so common in marine caves and is the dominant benthic species in the anchialine environment, the why were more sites and individuals analyzed? Is *P. misticia* really the most dominant species in the anchialine caves of the Ryukyus? The scope and impact of this work would be greatly increased by adding in more sampling sites. It might be beyond the scope of this study, but if you could include sites from the center of the distribution you could make comparisons regarding the population structure and gene flow at the center vs the margin of the distribution.

Material and Methods:

-More information regarding the sampling is warranted. In total, 48 individuals from three sampling sites were included in the MIG-seq analysis. Was this 16 individuals from each site or how was it divided up? Was that all that was collected or why were only 48 individuals analyzed?

-Please provide more detail on SNP filtering. The authors mention that "MIG-seq was used to amplify a few hundred to a few thousand genome SNPs", but they only end up analyzing 460 SNPs. Why were more SNPs not included.

-References in the "Population genetic analyses" (and throughout the manuscript) are improperly formatted. Each reference is given twice. Once as a number corresponding to the references list and once as the author and year of publication.

Results:

Please provide more details in the results section. You have generated a lot of data with the MIG-seq analysis, however not much detail is provided. Overall, how many SNPs were found? Were

there SNPs that were unique to a specific sampling site? The results of the SNP filtering? Can you use the SNPs to calculate effective population sizes? Etc.

-Please provide the results of the divMigrate analysis. You could provide the network graph or at least the relative values of gene flow between sampling sites.

-Did the authors look for F_{st} outlier SNPs?

Discussion:

In the discussion and throughout the manuscript, don't refer to the three sampling sites as "populations". You found that there was high gene flow and no structure between the three sampling sites, so they represent one panmictic population. Instead, refer to the three sampling sites as sampling sites.

Figure 1- Add a scale bar

Table 2- What is the ratio of the number of observed alleles? Why not report just the number of alleles observed?

Reviewer: 2

Comments to the Author(s)

This study examines gene flow in a marine cave-dwelling species of shrimp in the Ryukyu Islands in the Indo-West Pacific Ocean using next-generation SNP approach. The authors found no genetic differentiation or evidence of direction gene flow from three sites sampled, which contrasts other studies of cave-obligate shrimp species in the region. This is an interesting study; however, there are several comments as outlined below that I would like to see addressed before acceptance.

Given the audience of the journal, I feel that the authors should consider broadening the focus of the Introduction for a more general audience. As written in its current form, the manuscript is very marine cave specific. There is a large body of literature on geographic patterns of gene flow in marine and many other organisms, which can be reference to better place the study in a broader context.

Do you consider *P. misticia* a cave-obligate species?

Authors should consider a supplemental table describing field sampling sites in more detail.

Were specimens of *Parhippolyte misticia* identified morphologically? Or were sites that were sampled all from previously identified populations?

Why was such a low depth of coverage (~ 3) in your stacks analysis? If you increased to a minimum depth of 10, what effect would this have on your dataset and downstream analyses?

How did you assess allele dropout?

Did you test for violations of Hardy-Weinberg equilibrium and test of linkage between any SNPs?

You might also consider testing for population structure using approaches like ADMIXTURE or fastSTRUCTURE. You could also further explore population structure using DAPC and also demographic modeling and/or partial Mantel tests (there are issues with the latter) to explore isolation-by-distance.

Line 94 – You might consider briefly describing MIG-seq in detail for those unfamiliar with this approach.

Figure 1 could also use a scale bar.

Author's Response to Decision Letter for (RSOS-191731.R0)

See Appendix A.

RSOS-191731.R1 (Revision)

Review form: Reviewer 1

Is the manuscript scientifically sound in its present form?

Yes

Are the interpretations and conclusions justified by the results?

Yes

Is the language acceptable?

Yes

Do you have any ethical concerns with this paper?

No

Have you any concerns about statistical analyses in this paper?

No

Recommendation?

Accept with minor revision (please list in comments)

Comments to the Author(s)

Still don't see a scale bar in figure 1

Decision letter (RSOS-191731.R1)

Dear Dr Yorisue

On behalf of the Editors, we are pleased to inform you that your Manuscript RSOS-191731.R1 "Extensive gene flow among populations of the cavernicolous shrimp at the northernmost distribution margin in the Ryukyu Islands, Japan" has been accepted for publication in Royal Society Open Science subject to minor revision in accordance with the referees' reports. Please find the referees' comments along with any feedback from the Editors below my signature.

Please submit your revised manuscript and required files (see below) no later than 7 days from today's (ie 10-Sep-2020) date. Note: the ScholarOne system will 'lock' if submission of the revision is attempted 7 or more days after the deadline. If you do not think you will be able to meet this deadline please contact the editorial office immediately.

on behalf of the Associate Editor and Professor Kevin Padian (Subject Editor)
openscience@royalsociety.org

Associate Editor comments to Author:

Thank you for submitting your revised work to us. Your manuscript has now been reviewed by one of the original referees who is now satisfied with the revisions made, however we note that a scale bar is still missing from Figure 1. Please address this. Additionally, before resubmitting, I note a number of typos and English language errors which should be rectified -- please contact a professional language editing service (<https://royalsociety.org/journals/authors/language-polishing/>) prior to resubmitting the manuscript. Alternatively, you may consider asking a native speaker of English for their guidance. If requested to edit the written English, you must provide proof that you have done so: acceptable proof includes a certificate of language-editing from a language editing service or a signed letter from a native speaker of English.

Finally, please note the open data and code policy for Royal Society Open Science: <https://royalsociety.org/journals/authors/author-guidelines/#data> This includes all raw additional data, analysis code, or code used to run modelling or analyses. Please ensure that any additional code and data files are either provided as 'electronic supplementary material' files to your revision, or uploaded to a repository such as Dryad

Reviewer comments to Author:
Reviewer: 1
Comments to the Author(s)

Still don't see a scale bar in figure 1

===PREPARING YOUR MANUSCRIPT===

===PREPARING YOUR REVISION IN SCHOLARONE===

- Ensure that your data access statement meets the requirements at <https://royalsociety.org/journals/authors/author-guidelines/#data>. You should ensure that you cite the dataset in your reference list. If you have deposited data etc in the Dryad repository, please only include the 'For publication' link at this stage. You should remove the 'For review' link.
- If you are requesting an article processing charge waiver, you must select the relevant waiver option (if requesting a discretionary waiver, the form should have been uploaded at Step 3 'File upload' above).
- If you have uploaded ESM files, please ensure you follow the guidance at <https://royalsociety.org/journals/authors/author-guidelines/#supplementary-material> to include a suitable title and informative caption. An example of appropriate titling and captioning may be found at https://figshare.com/articles/Table_S2_from_Is_there_a_trade-off_between_peak_performance_and_performance_breadth_across_temperatures_for_aerobic_scooping_in_teleost_fishes_/3843624.

Author's Response to Decision Letter for (RSOS-191731.R1)

See Appendix B.

Decision letter (RSOS-191731.R2)

Dear Dr Yorisue,

It is a pleasure to accept your manuscript entitled "Extensive gene flow among populations of the cavernicolous shrimp at the northernmost distribution margin in the Ryukyu Islands, Japan" in its current form for publication in Royal Society Open Science.

Due to rapid publication and an extremely tight schedule, if comments are not received, your paper may experience a delay in publication. Royal Society Open Science operates under a continuous publication model. Your article will be published straight into the next open issue and this will be the final version of the paper. As such, it can be cited immediately by other

researchers. As the issue version of your paper will be the only version to be published I would advise you to check your proofs thoroughly as changes cannot be made once the paper is published.

on behalf of Professor Kevin Padian (Subject Editor)
openscience@royalsociety.org

Appendix A

Dear Editor of Royal Society Open Science,

We appreciate your handling our manuscript submitted to Biofouling. Both reviewers 1 and 2 recommended major revision before publication. We have basically followed them. Reviewer 2 suggested to use STRUCTURE approach and we added the analysis. In addition, reviewer 2 suggested to consider the SNP filtering methods, and we have changed the setting of depth coverage in stacks analysis, which resulted in 62 SNPs available for the down-stream analyses.

Our revision includes re-writing of text, and changes on figures and tables. However, results and conclusions are basically unchanged. Below, we provide the point-by-point response to the reviewers' comments.

Reviewer 1

The manuscript by Yorisue et al., attempts to investigate patterns of gene flow of the anchialine shrimp *Parhippolyte misticia* at its distribution margin using high throughput sequencing technologies (i.e. ISSR genotyping). The authors sampled 3 locations and found no population structure and evidence of high geneflow between all sampling locations. While I found the subject matter to be interesting, I feel that the authors failed to provide enough details to justify the merit of their work. The impact of this study could be greatly increased if more sampling site were included. However, if that is beyond the means of this study, providing more detail throughout the manuscript would improve it greatly.

Reply

While I believe this manuscript would benefit from heavy revisions before an in-depth review, some of my major comments are outlined below.

We appreciate the positive feedback on our manuscript. We address each comment below.

Comment 1

Line 47- delete "therefore"

Reply to comment 1

Deleted "therefore" as suggested.

Comment 2

Not sure I would agree with Line 58-59. mtDNA can be useful in examining larval dispersal potential, but agree MIG-seq should provide much better resolution.

Reply to comment 2

We agree to the reviewer's point and deleted the sentence.

Comment 3

If *P. misticia* is so common in marine caves and is the dominant benthic species in the anchialine environment, the why were more sites and individuals analyzed? Is *P. misticia* really the most dominant species in the anchialine caves of the Ryukyus? The scope and impact of this work would be greatly increased by adding in more sampling sites. It might be beyond the scope of this study, but if you could include sites from the center of the distribution you could make comparisons regarding the population structure and gene flow at the center vs the margin of the distribution.

Reply to comment3

Actually *P. misticia* is a dominant species in marine caves in the Ryukyu Islands, but we agree to the point that it is not always "the most dominant". We revised the sentence as followings.

"In the Ryukyu Islands, *P. misticia* is a dominant benthic species in marine caves."
(L.80-81)

We agree that the analyses on more sites and individuals increase the impact of this study. However, we do not have the samples from any other sites and including additional samples from other sites is beyond this study.

Comment 4

More information regarding the sampling is warranted. In total, 48 individuals from three sampling sites were included in the MIG-seq analysis. Was this 16 individuals from each site or how was it divided up? Was that all that was collected or why were only 48 individuals analyzed?

Reply to comment4

We described the number of individuals analysed from each site in materials and methods (L.90-91). To minimize the effect of field survey on the environment, we did not collect too much samples. Actually, we analysed 68 individuals in total, in which

each 24 was from Hedo and Ie, and 20 were from Shimoji. In the original manuscript we used the data from 48 individuals that have passed the SNP filtering process. In the revised manuscript, we changed the condition of SNP filtering (Please see reply to comment 5) and we could use the data from 68 individuals. We believe analyses on these 68 individuals gives information on population genetic structure, which is worth publishing.

Comment 5

Please provide more detail on SNP filtering. The authors mention that “MIG-seq was used to amplify a few hundred to a few thousand genome SNPs”, but they only end up analyzing 460 SNPs. Why were more SNPs not included.

Reply to comment 5

We described the total number of SNPs detected, and the number of SNPs used for the analyses (L.126-128). In the revised manuscript, we analysed SNPs from four different filtering methods (Please see Reply to comment 6 from reviewer 2). We could not use most SNPs that were trapped at the filtering steps.

Comment 6

References in the “Population genetic analyses” (and throughout the manuscript) are improperly formatted. Each reference is given twice. Once as a number corresponding to the references list and once as the author and year of publication.

Reply to comment 6

We checked and formatted references through the manuscript.

Comment 7

Please provide more details in the results section. You have generated a lot of data with the MIG-seq analysis, however not much detail is provided. Overall, how many SNPs were found? Were there SNPs that were unique to a specific sampling site? The results of the SNP filtering? Can you use the SNPs to calculate effective population sizes? Etc.

Reply to comment 7

We added the results of the SNP filtering in the result section (L.149-151). It is possible to estimate the relative effective population size. We agree that it is interesting to estimate and compare the effective population size if we could analyse more samples

from other sites from center of the distribution in the future, but it is beyond the current study.

Comment 8

Please provide the results of the divMigrate analysis. You could provide the network graph or at least the relative values of gene flow between sampling sites.

Reply to comment 8

We added the Fig3 that showing the result of divMIGRATE.

Comment 9

Did the authors look for Fst outlier SNPs?

Reply to comment 9

We checked the outlier SNPs by Bayescan and we detected no outlier (L.150-151).

Comment 10

In the discussion and throughout the manuscript, don't refer to the three sampling sites as "populations". You found that there was high gene flow and no structure between the three sampling sites, so they represent one panmictic population. Instead, refer to the three sampling sites as sampling sites.

Reply to comment 10

We revised to refer the three sampling sites as sampling sites as suggested.

Comment 11

Figure 1- Add a scale bar

Reply to comment 11

We added the scale bar in Figure 1.

Comment 12

Table 2- What is the ratio of the number of observed alleles? Why not report just the number of alleles observed?

Reply to comment 12

We revised table 2 to report number of alleles observed, not ratio of the number of observed alleles.

Reviewer 2

This study examines gene flow in a marine cave-dwelling species of shrimp in the Ryukyu Islands in the Indo-West Pacific Ocean using next-generation SNP approach. The authors found no genetic differentiation or evidence of direction gene flow from three sites sampled, which contrasts other studies of cave-obligate shrimp species in the region. This is an interesting study; however, there are several comments as outlined below that I would like to see addressed before acceptance. Given the audience of the journal, I feel that the authors should consider broadening the focus of the Introduction for a more general audience. As written in its current form, the manuscript is very marine cave specific. There is a large body of literature on geographic patterns of gene flow in marine and many other organisms, which can be reference to better place the study in a broader context.

Reply

We appreciate reviewer 2's positive feedback on our research. In the revised manuscript, we added new paragraph in introduction that describe the importance of research on gene flow of coastal organisms. We address each comment below.

Comment 1

Given the audience of the journal, I feel that the authors should consider broadening the focus of the Introduction for a more general audience. As written in its current form, the manuscript is very marine cave specific. There is a large body of literature on geographic patterns of gene flow in marine and many other organisms, which can be reference to better place the study in a broader context.

Reply to comment 1

We added new paragraph in Introduction that describe the importance with context of conservation to understand gene flow pattern of coastal species. (L.45-48)

Comment 2

Do you consider *P. misticia* a cave-obligate species?

Reply to comment 2

P. misticia distributes only in marine caves during day time, but we observed that it distributes depression on coral reefs during night time. Therefore, we have decided not to use “cave-obligate species” in this manuscript.

Comemnt 3

Authors should consider a supplemental table describing field sampling sites in more detail.

Reply to comment 3

We added the latitude/longitude of sampling sites in materials & methods (L. 94-95).

Comment 4

Were specimens of *Parhippolyte misticia* identified morphologically? Or were sites that were sampled all from previously identified populations?

Reply to comment 4

We identified the specimens based on morphological characters. We added the statements in (L. 92).

Comment 5

Why was such a low depth of coverage (-m 3) in your stacks analysis? If you increased to a minimum depth of 10, what effect would this have on your dataset and downstream analyses?

Reply to comment 6

To evaluate the effect of depth setting on the downstream analyses, we compare the STRUCTURE results of SNPs set derived from different depth settings (-m 3 amd -m 10). Both analyses indicated that $K = 2$ best explained the data and no genetic differences corresponded with the three sampling sites. However, percentages of admixture for each individual were different between the two analyses. In the revised manuscript, we show the results of analyses with -m 10 because it is commonly used in this kind of studies.

Comment 7

How did you assess allele dropout?

Reply to comment 7

In stacks, we used loci that were detected from at least 70 of the individuals analysed (i.e., r was set at 70%). We described about r setting in L. 118-120.

Comment 8

Did you test for violations of Hardy-Weinberg equilibrium and test of linkage between any SNPs?

Reply to comment 8

We have tested the existence of F_{st} outlier by BayeScan to detect possible SNPs under natural selection, but we detected no outlier SNP. (L.128-289, 150-151)

Comment 9

You might also consider testing for population structure using approaches like ADMIXTURE or fastSTRUCTURE. You could also further explore population structure using DAPC and also demographic modeling and/or partial Mantel tests (there are issues with the latter) to explore isolation-by-distance.

Reply to comment 9

We conducted STRUCTURE analysis to further understand details of population structure of *Parhippolyte misticia*. The result is shown in Fig. 2. Because we analysed samples from only there sites, we did not conduct the mantel test.

Comment 10

You might consider briefly describing MIG-seq in detail for those unfamiliar with this approach

Reply to comment 10

We already described about MIG-seq approach in L. 98-103 and we think it is enough to explain about MIG-seq.

Comment 11

Figure 1 could also use a scale bar.

Reply to comment 11

We added the scale bar in Figure 1.

Appendix B

Dear Editor of Royal Society Open Science,

We appreciate your handling our manuscript (RSOS-191731.R1) submitted to Royal Society Open Science. Our manuscript was accepted with minor revision. Below, we provide the response to the Editor and reviewer comments.

Associate editor

Please address this. Additionally, before resubmitting, I note a number of typos and English language errors which should be rectified

Response

English of the original manuscript was revised by English editing service (<http://www.elss.co.jp/en/>). Please find the Email from ELSS. We have carefully checked the typos of the final manuscript.

Reviewer: 1

Still don't see a scale bar in figure 1

Response

We added the scale in figure 1.